# Effects of Probiotics in Adults with Gastroenteritis: A Systematic Review and Meta-Analysis of Clinical Trials

**DOI:** 10.3390/diseases11040138

**Published:** 2023-10-11

**Authors:** Amal K. Mitra, Adetoun F. Asala, Shelia Malone, Malay Kanti Mridha

**Affiliations:** 1Department of Epidemiology and Biostatistics, College of Health Sciences, Jackson State University, Jackson, MS 39213, USA; adetoun.asala1@msdh.ms.gov (A.F.A.); shelia.malone@students.jsums.edu (S.M.); 2Office of Preventive Heath, Mississippi State Department of Health, Ridgeland, MS 39157, USA; 3Brac James P. Grant School of Public Health, Center for Non-Communicable Disease and Nutrition, Brac University, Dhaka 1213, Bangladesh; malay.mridha@bracu.ac.bd

**Keywords:** probiotics, clinical trials, adults, gastroenteritis, inflammatory bowel disease, PRISMA

## Abstract

Probiotics have been widely used in gastroenteritis due to acute and chronic illnesses. However, evidence supporting the effectiveness of probiotics in different health conditions is inconclusive and conflicting. The aim of this study was to review the existing literature on the effects of probiotics on gastroenteritis among adults. Only original articles on clinical trials that demonstrated the effects of probiotics in adults with gastroenteritis were used for this analysis. Multiple databases, such as PubMed, Google Scholar, MEDLINE and Scopus databases, were searched for the data. The study followed standard procedures for data extraction using a PRISMA flow chart. A quality appraisal of the selected studies was conducted using CADIMA. Finally, a meta-analysis was performed. Thirty-five articles met the selection criteria; of them, probiotics were found effective in the treatment and/or prevention of chronic inflammatory bowel disease (IBD), including ulcerative colitis and Crohn’s disease in 17 (49%), and the treatment of *pouchitis* in 4 (11.4%), antibiotic-induced diarrhea in 3 (8.6%), *Helicobacter pylori* infection in 2 (5.7%) and diverticulitis in 1 (2.9%), while the remaining 7 (20%) were ineffective, and 1 study’s results were inconclusive. The meta-analysis did not demonstrate any significant protective effects of probiotics. Having a τ^2^ value of zero and I^2^ of 6%, the studies were homogeneous and had minimum variances. Further studies are suggested to evaluate the beneficial effects of probiotics in IBDs and other chronic bowel diseases.

## 1. Introduction

Gastroenteritis poses serious public health concerns in both high- and low-income countries. It is one of the leading causes of morbidity and mortality worldwide [1]. Globally, the estimated annual cost of healthcare and loss of productivity due to gastroenteritis is about USD 60 billion, of which developing countries bear the highest burden [1,2]. According to the Centers for Disease Control and Prevention (CDC), gastroenteritis is the most prevalent infectious disease syndrome in the United States, accounting for over 350 million illnesses annually and about 200,000 deaths, with the elderly having higher mortality risks [3,4]. The symptoms of gastroenteritis can range from mild asymptomatic infections to life-threatening conditions and death [5]. Deaths from acute causes of gastroenteritis occur as a result of profound dehydration [6]. Inflammatory bowel diseases (IBDs), such as underactive colitis (UC) and Crohn’s disease (CD), are important causes of gastroenteritis. IBDs may result in an increased lifetime risk of serious complications or manifestations, including fistula or abscesses, strictures, diverticulitis, gastrointestinal bleeding, toxic megacolon, perforation of the bowel, ischemic colitis, drug-induced colitis and perianal fistula [7].

Probiotics are supplements or foods that contain live non-pathogenic microorganisms, which can maintain and improve microbial balance in the gastrointestinal tract [8]. Some beneficial effects of probiotics relevant to the treatment and prevention of gastroenteritis include the following: reduction in invasion and colonization of the intestine by pathogenic organisms, modification of host immune response, and reduction in pH in the intestine [8,9]. Although, some studies confirmed that probiotics have anti-inflammatory and antimicrobial effects and help maintain good bacteria in the gut, results from some other studies are inconclusive and conflicting [10].

A systematic review proved that the use of probiotics was effective in reducing the duration of acute diarrhea by 14% in children, but the number of studies was limited [11]. Another study with a meta-analysis showed that probiotics are effective in reducing the duration of acute diarrhea in children by 26% [12]. However, there was no effect of probiotics in reducing the risk of hospitalization. On the other hand, studies of probiotics in adults with acute diarrhea yielded mixed results—a review of two clinical trials of the use of enterococcus SF 68 was able to shorten the duration of diarrhea [13], whereas another double-blind controlled clinical trial regarding the use of *Streptococcus faecium* SF 68 found that it was ineffective in adults with acute diarrhea [14]. In general, probiotics were proven useful in reducing antibiotic-induced diarrhea [15] and irritable bowel syndrome [16] but they had little or no effect for the maintenance of remission from ulcerative colitis (UC) [17] and Crohn’s disease (CD) [18], as evidenced from a systematic review and meta-analysis.

Based on the available data, the use of specific probiotic preparations should be evaluated cautiously using evidence from well-designed clinical trials. Therefore, the purpose of this study was to present the results of a systematic review and meta-analysis carried out to examine the effects of probiotics in adults with acute and chronic gastroenteritis of multiple etiologies.

## 2. Materials and Methods

A systematic literature search was conducted from February to May 2021 using PubMed database as the primary data source. Other research databases included were Google Scholar, MEDLINE and Scopus. The search keywords in PubMed were “effects of probiotics in gastroenteritis”; Medical Subject Headings (MeSH) Terms were: “effect*” OR “outcomes” OR “impact” OR “efficacy*” OR “efficaciousness*” AND “Probiotics” AND “Gastroenteritis” AND “Clinical Trial”.

Articles that met the following criteria were included in the review: (1) studies published between the years 1990 and 2022; (2) only clinical trial study designs for more consistency in the results; (3) studies related to the effects of probiotics in gastroenteritis; any other diseases where probiotics were used were not considered in this study; (4) studies with adult participants over 19 years of age; (5) full-text articles; and (6) articles written in English language. Figure 1 shows the detailed search strategies for this study using the preferred reporting items for systematic reviews and meta-analyses (PRISMA, Berlin, Germany) [19]. The protocol for the systematic review of our study was not registered for PROSPERO.

### Quality Appraisal Methods

Studies were appraised for quality using CADIMA [20]. CADIMA is a free tool, which is available online for managing articles for systematic reviews, including automated duplicate removal, uploading PDF articles and documentation of the review process. CADIMA facilitates the conduct and documentation of systematic reviews, systematic maps and literature reviews. CADIMA helps in automatic duplicate article removal and allows for a detailed documentation of the review process. Through CADIMA, standards for critical appraisal and the rating scale were defined. We followed the essential tools of appraisal for systematic reviews developed by the University of Adelaide, South Australia [20]. A rating scale from 0 to 4 was based on the following criteria: (1) sample size: greater than 30 = 1; smaller sample = 0; (2) randomized controlled trials = 1; not randomized, no controls = 0; (3) studied both safety and efficacy = 1; otherwise = 0; (4) standard and objective evaluation criteria = 1; otherwise = 0. Based on the criteria mentioned earlier, we rated each of the 35 studies independently in a range of 0 to 4.

## 3. Results

Of the 1865 research articles identified initially through a database search, 1544 articles were excluded for not being conducted in adults. Of the remaining 321 studies, studies excluded with reasons were as follows: 190 studies did not involve clinical trials; 84 were not full articles; and 12 did not meet the eligibility criteria and were not related to gastroenteritis. The results presented below include analyses of 35 studies for the systematic review (Figure 1).

The total sample size for the 35 studies was 4577, ranging from 15 to 777 samples in individual studies, with a median of 44. Only 12 studies (34%) had a sample size of more than 100.

### 3.1. Type of Illnesses Associated with Diarrhea

All 35 studies reviewed here are presented with gastroenteritis. All of them had chronic diarrhea of diverse etiologies (such as IBD, pauchitis, antibiotic-associated diarrhea, etc.), except one [14], who had acute watery diarrhea (Table 1 and Figure 2). We found that the majority of the studies (51%, 18 of 35) focused on the effectiveness of probiotics in the treatment of IBDs (UC and CD), 11% (4 of 35) of the study patients had pouchitis (inflammation that occurs in the lining of a pouch created during surgery to treat ulcerative colitis or certain other diseases), about 9% (3 of 35) had antibiotic-induced diarrhea, 6% (2 of 35) had diarrhea due to *Helicobacter pylori*, and 1 each (2.9%) had diverticulitis or acute watery diarrhea due to *Vibrio cholerae* and enterotoxigenic *Escherichia coli* infection (Figure 2).

Detailed information about the 35 selected studies [11,12,13,14,15,16,17,18,19,20,21,22,23,24,25,26,27,28,29,30,31,32,33,34,35,36,37,38,39,40,41,42,43,44,45], the type of gastroenteritis studied, type of probiotics used and their effectiveness is presented in Table 1.

### 3.2. Probiotic Strains Used and Follow-Up

Most studies (60%, *n* = 21) administered multiple strains of probiotics, while the remaining 14 administered single strains of probiotics. The most commonly administered probiotic strains were *Lactobacilli*, *Bifidobacteria*, *Streptococcus* and *Escherichia*. In a few studies, probiotics were administered as an adjuvant therapy with another conventional treatment, such as an anti-inflammatory drug for IBDs (such as pentasa or mesalazine [25,30,41]), rehydration therapy for acute watery diarrhea [14] or with a combination of treatments comprising antibiotics and proton pump inhibitors (PPIs) (Esomeprazole) for chronic gastritis or duodenal ulcers due to *H. pylori* infection [52].

Patients’ follow-up protocols varied widely, ranging from as low as 10 days to 2 years depending on the illness type. Thus, 10 studies followed patients for 12 months or more, 12 followed patients for 3–11 months and another 14 followed patients for 3 months or less.

### 3.3. Quality Appraisal Findings

Due to having no significant inter-observer variations in evaluating the quality of the studies, an average of four scores is presented in Table 1. Out of 35 studies reviewed, 19 (55%) scored highly (4 out of 4), 15 (43%) scored moderately (3 out of 4) and only 1 was rated as poor (2 out of 4). Among the 27 studies that were proven to be effective for the treatment of probiotics, the majority (63%, *n* = 14) were of high quality (score 4 out of 4), 44% (*n* = 12) scored moderate quality (score 3 out of 4) and only 1 (4%) scored a poor rating (score 2 out of 4).

### 3.4. Efficacy and Safety of Probiotics

The outcome measures were considered favorable if studies reported resolution, remission, improvement or no relapse of gastroenteritis after treatment. Of the 35 studies reviewed, 27 (77%) showed a favorable response after using probiotics, 7 (20%) indicated that probiotics were ineffective, and 1 study conducted in Iran [43] was inconclusive (Table 1). Probiotics were most effective in treating or preventing gastroenteritis due to IBDs (Figure 2). Of the various forms of probiotics, a mixture of multiple probiotic strains, in the form of *VSL* #3, was found to be most effective in treating patients with IBDs [32,39,48,49]. In three studies [21,40,44], gastroenteritis was induced by antibiotics [21,44] or non-steroidal anti-inflammatory drugs [44]—of them, VSL #3 was used in two studies for the relief of gastrointestinal symptoms [40,44], and *lactobacillus casei* was used in one [21] for the prevention of gastroenteritis. The antibiotics that were reported to be most frequently associated with diarrhea were as follows: cephalosporins, penicillins and fluoroquinolones [21]. Only one study [21] out of three reported antibiotic-associated gastroenteritis among those who were treated with probiotics and controls. Antibiotic-associated diarrhea was documented in fewer subjects after giving probiotic drinks (12.5% vs. 31.3%, between probiotic group and control, respectively), suggesting that there is efficacy in probiotics in preventing gastroenteritis after antibiotic use. However, probiotics were found to be ineffective in five studies in patients with UC or CD [22,24,27,36,37] and also in two other studies—one on 20 patients with pouchitis in Finland [33] and another study on 183 patients with a severe form of acute watery diarrhea due to *V. cholerae* and *E. coli* infections in Bangladesh [14] (Table 1).

All probiotics administered in these studies, including seven studies [14,22,24,27,33,36,37] that were proven ineffective (as mentioned earlier), were well tolerated by patients, and no adverse side effects were reported. However, several studies cautioned the use of probiotics among immunocompromised patients due to safety concerns in such patients [10].

### 3.5. Effectiveness of Probiotics, as Evaluated through Meta-Analysis of 22 Clinical Trials

Due to the unavailability of relevant data, a meta-analysis was conducted using 22 out of 35 (63%) studies. Table 2 shows the relative risk and 95% confidence intervals (CIs) of the effect of probiotics in each study. Risk ratios demonstrated a protective effect in 50% of the studies (*n* = 11); however, 95% CIs included 1 in each of them. The pooled relative risk was 0.99, with 95% CI being 0.90 and 1.09. A test for the overall effect showed a *p*-value of 0.37, meaning that there was not enough evidence to indicate that the intervention had a significantly more protective effect than controls. The value of τ^2^ indicated low variation in true effects. The Higgins H test (I^2^) was 6%, indicating a homogeneous nature of the weights of the studies evaluated in the meta-analysis. It is important to note that although the clinical trials reviewed here were homogenous in nature, they were widely diverse in terms of etiologies and the type of probiotics used.

In Figure 3, a forest plot, risk ratio and 95% CI included the line of no effect, and the *p*-value for the overall effect was 0.85. Because of negligible heterogeneity among the studies, we can rely on the aggregated estimate more as the majority or all individual studies reached the same conclusion.

### 3.6. Overall Risk of Bias by Categories of Bias

As shown in Figure 4 and Figure 5, the domains evaluated in this review were as follows: (a) Random Sequence Generation: it is a form of selection bias (biased allocation to interventions), resulting from insufficient generation of a randomized sequence. (b) Allocation Concealment: it is also a type of selection bias (biased allocation to interventions), owing to insufficient allocation concealment prior to assignment. (c) Blinding of Participants and Personnel: this is a performance bias due to participants’ and personnel’s knowledge of the assigned interventions during the study. (d) Blinding of Outcome Assessment: this is a detection bias due to the outcome assessors’ knowledge of the allocated interventions. (e) Incomplete Outcome Data: this is an attrition bias due to the quantity, character or treatment of incomplete outcome data. (f) Selective Reporting: this is bias in reporting, owing to the selective reporting of outcomes. (g) Other Bias: this category encompasses biases caused by issues not covered elsewhere in the table. In Figure 4 and Figure 5, the rating scales of bias are high (red), low (green), and unclear (yellow).

Based on Figure 5, only one study [45] (Steed et al., 2010) was judged as having a high risk of bias for two domains: (1) Random Sequence Generation (selection bias)—reviewer’s comments: there was no description of the randomization process and the domain selection reporting (reporting bias). (2) Incomplete Outcome Data (attrition bias)—reviewer’s comments: the handling of incomplete outcome data was not described in detail.

### 3.7. Assessment of Publication Bias

Figure 6 shows that the larger studies cluster around the top of the plot, and smaller studies are spread across the bottom. This is an ideal funnel plot where the included studies are scattered on either side of the overall effect line symmetrically. There is no severe asymmetry to either side, so we conclude that publication bias was not present.

### 3.8. Sub-Group Meta-Analysis for Studies Reporting Ulcerative Colitis Only

Among the reviewed studies, the majority (23 out of 35, 66%) had IBDs. Of the patients with IBDs, 12 had UC only; of them, 1 was ineffective and 11 were found effective for the treatment with probiotics in our systematic review. In this cohort of patients with IBDs, eight had both UC and CD and two had CD as a single disease entity diagnosed (Table 1). We conducted meta-analysis on a subgroup of studies that reported UC only. Three of the studies [29,50,54] were not included in the analysis because they did not include a control group in the protocols. The meta-analysis results of sub-group analysis are presented in Figure 7.

Even though the systematic review suggests that probiotics are effective, the overall effect of the meta-analysis of studies with UC patients indicates that adverse events caused by probiotics outweigh the benefits, and probiotics were not effective in UC (*p* = 0.28).

## 4. Discussion

### 4.1. Effective Treatment with Probiotics

Out of 35 studies that were investigated for the systematic review, our meta-analysis included 22 (63%) studies. The systematic review and meta-analysis results demonstrated a mixed effect of probiotics therapy in gastroenteritis in adults. However, studies found that either short-term or long-term administration of probiotics is safe.

Our analysis suggests that based on the systematic review per se, probiotics were an effective therapeutic alternative or an adjuvant therapy for gastroenteritis in 27 out of 35 (77%) of studies. Our systematic review was in congruence with systematic reviews of probiotics in adults with viral gastroenteritis [55] and IBDs (mostly in ulcerative colitis) [56]. However, several recent systematic reviews showed no or minimum beneficial effects of probiotics in Crohn’s disease [56,57]. Some study results showed mixed effects in terms of having benefits of using probiotics in reducing the risk of gastrointestinal infections (RR 0.86, 05% CI 0.73 to 1.101) but having no effects on the duration or severity of gastrointestinal infection. The authors cautioned use of the results due to a low number of studies, high risk of bias and heterogeneity in the studies [58]. The study results may be not comparable, partly because of differing outcome measures [58] and also not having results of a meta-analysis in all reported studies [56,57]. Further studies are suggested for understanding the effect of probiotics as a single therapy or in combination with standard treatments for Crohn’s disease and gastroenteritis due to other etiologies.

In contrast to the systematic review results, the meta-analysis of the aggregated data did not provide enough evidence to support statistically significant protective effects of probiotics in the study subjects with gastroenteritis. The systematic analysis also showed that probiotics were not effective in seven (20%) patients with gastroenteritis. The diverse nature of etiologies of gastroenteritis (such as IBDs, pauchitis, antibiotic-associated diarrhea, etc.) in the combined group meta-analysis of 22 studies could result in varying degrees of diseases severity and duration of symptoms, which could partly explain the negative results of the meta-analysis.

To further explore the reasons behind the observed discrepancies between the findings of the systematic review and meta-analysis, we conducted a subgroup meta-analysis of studies on UC only. Interestingly, the overall meta-analysis findings of studies having gastroenteritis due to UC were in alignment with the total group meta-analysis results, suggesting that probiotics were found ineffective in either situation. However, the study sample of our subgroup study was only eight studies comprising 715 adult patients with UC. Therefore, larger disease-specific studies would add further value.

Disease conditions that improved upon probiotic treatment mostly included IBDs due to UC. Other disease entities were chronic gastritis, pauchitis, *H. pylori* infections and drug-induced enteropathies. No particular disease entities were found ineffective in the treatment with probiotics. In this review, the combined therapy of probiotics with nonsteroidal anti-inflammatory drugs, proton pump inhibitors or antibiotics showed effective results in treating IBDs, chronic gastritis and infection due to *H. pylori*. Probiotics were also effective in the disease prevention or prevention of relapses in eight studies [21,31,35,36,37,47,52,54]. The findings of this review also suggested that probiotics were useful in improving patients’ quality of life in two studies [39,51], in addition to reducing morbidity due to chronic gastroenteritis.

Our systematic review data are consistent with earlier reports, confirming probiotics’ effectiveness in gastroenteritis. However, probiotics’ effectiveness depends on dose, strains used in probiotics, duration of therapy and the type of illnesses [9,59]. One may compare the effects of probiotics in adults with the ones described in studies in children, primarily to evaluate the effectiveness of probiotics on gastroenteritis due to different etiologies [60].

### 4.2. Ineffectiveness of Probiotics

Notably, approximately 20% (*n* = 7) of the studies reported that probiotics were ineffective in treating gastroenteritis, which creates an interesting discrepancy. The reasons for an ineffective outcome of these studies are mostly speculative at this point. The study of acute watery diarrhea due to *V. cholerae* and *E. coli* infections in Bangladesh [14] was more severe in nature than gastroenteritis due to other causes, and these patients with acute watery diarrhea needed intravenous rehydration. In such severe forms of diarrheal diseases, the effectiveness of probiotics remains a question. However, in addition to the results of using a lyophilized form of *Streptococcus feacium* SF68, further studies may be evaluated by using other forms of probiotics or using a combination of probiotics and antibiotics in acute diarrhea. In the other failure cases, all happened to be in patients with IBDs. Our sub-group analysis also showed a similar direction of no beneficial effects of probiotics in adults with ulcerative colitis. Matsuoka et al. (2018) [37] emphasized that the lack of effectiveness may be due to an inadequate dose of administered probiotics, the route of administration or the inability to confirm improvement by using endoscopic and/or imaging procedures. Another report showing no improvement after a symbiotic therapy in patients with mild-to-moderate IBDs could be attributable to a small sample size and the absence of more specific and objective biomarkers of inflammation, such as fecal calprotectin and histologic scores, to diagnose the disease [22].

### 4.3. Safety Issues

Our review did not find any safety concerns of probiotics in any of the studies. However, a previous study reported an increased risk of gastrointestinal symptoms, including abdominal pain in IBD patients taking probiotics, compared than those exposed to placebo (RR 2.59, 95% CI 1.28 to 5.22) [61]. Several studies described and warned about major safety issues in using probiotics, including, but not limited to, systemic infections, deleterious metabolic activities, excessive immune stimulation in susceptible individuals, gene transfer and gastrointestinal side effects [62,63]. Studies also cautioned that the administration of probiotics among vulnerable populations, especially immunocompromised individuals, should be carefully considered [10,64]. However, a solution to this conundrum may lie in the idea of making the report of adverse events involving mandatory and standardized probiotics, thereby improving product safety and data reliability [65].

### 4.4. Limitations of the Study

Our study cannot be generalized because only studies published in the English language and those having free full texts were reviewed. However, the nature of a comprehensive systematic review, using 30 years as the timeframe for the published reports, and inclusion of study findings of randomized controlled clinical trials only may limit the possibilities of bias. In addition, risks of bias assessments were low in all studies except one in our meta-analysis. A very low value (6%) on the Higgins test (I^2^) of heterogeneity also indicates the homogeneous nature of the evaluated studies. However, the results could be affected by the diverse nature of disease etiologies and possible diversity in disease severity and duration of illnesses in the analyzed studies. Another potential limitation is that we did not have access to individual-level data. We would probably have a better idea about the effectiveness of probiotics if we could analyze individual-level data instead of aggregated data reported in the studies.

## 5. Conclusions

Our meta-analysis provided new information in contrast to the data obtained from the systematic review on the effectiveness of probiotics in gastroenteritis in adults. Even the data gathered from the systematic review had a mixed effect of probiotics—although probiotics were effective in treating and reducing relapses of chronic inflammatory gastrointestinal conditions in most adults (78%), the review shows that probiotics are ineffective in about 20% of patients. More importantly, the pooled data of the meta-analysis demonstrated no statistically significant protective effects of using probiotics. However, there was a paucity of data from developing countries. Because of the observed differences in the meta-analysis results between studies, there is a need for research efforts to identify the most appropriate use of probiotics in various etiologies of gastroenteritis. Further studies are also needed to confirm whether probiotics can restore the gut microflora and improve gastroenteritis as a single therapy or adjunct therapy with other conventional treatments for the infection.

## Figures and Tables

**Figure 1 diseases-11-00138-f001:**
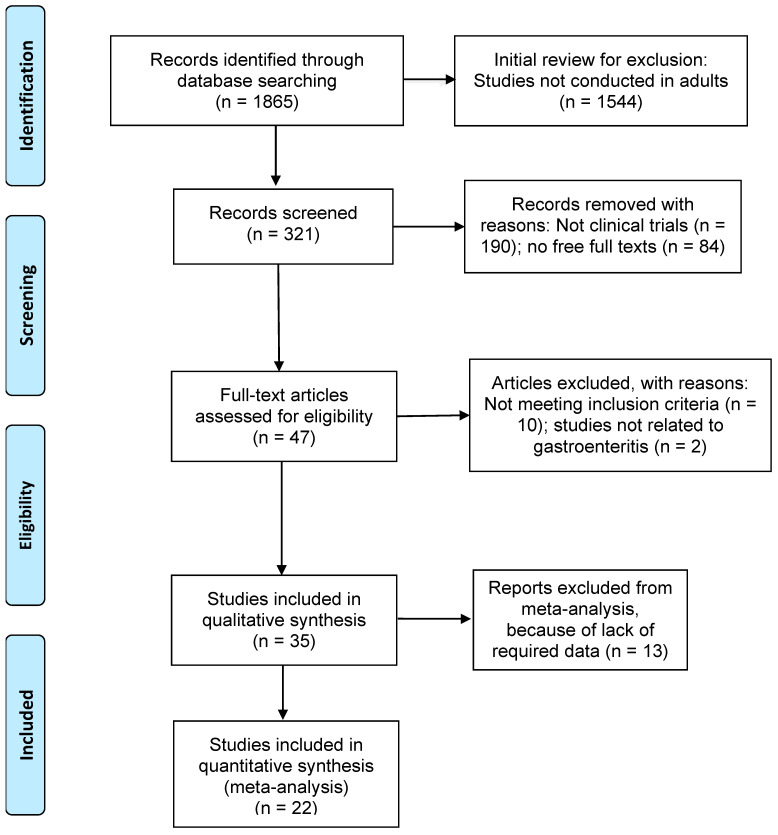
PRISMA flow chart showing inclusion and exclusion of studies.

**Figure 2 diseases-11-00138-f002:**
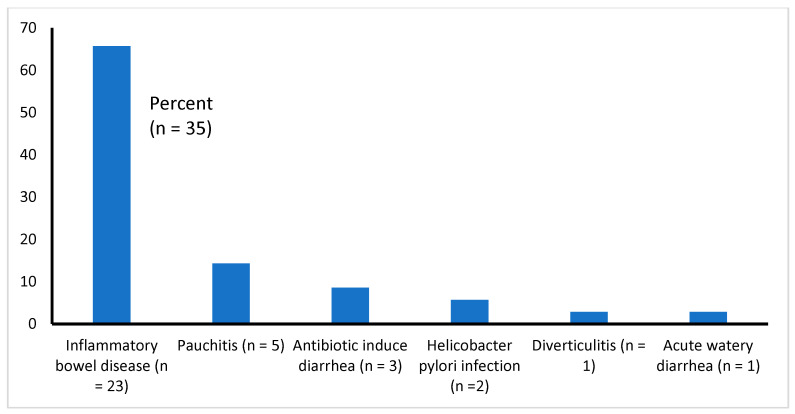
Types of infections causing diarrhea among the study samples (*n* = 35).

**Figure 3 diseases-11-00138-f003:**
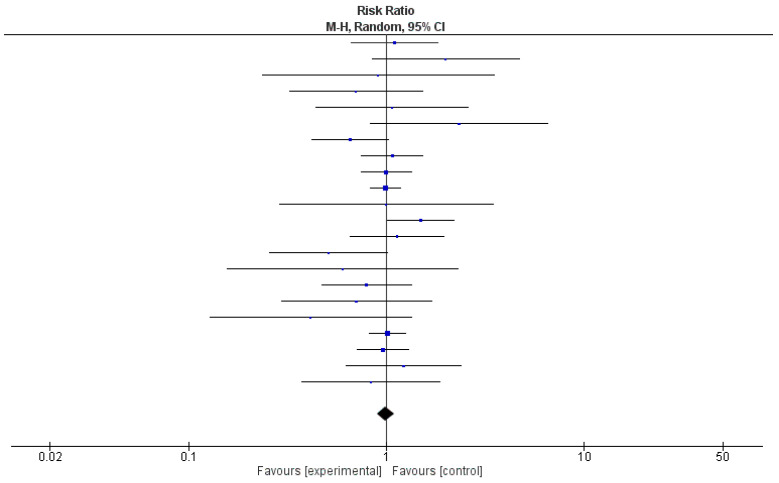
Forest plot for the pooled analysis of 22 included studies.

**Figure 4 diseases-11-00138-f004:**
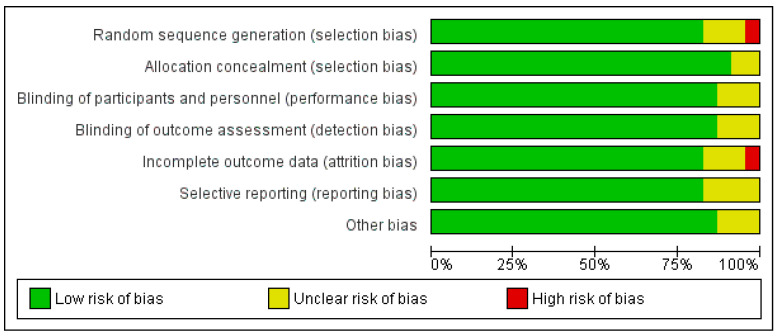
Risk of bias graph: review authors’ judgements about each risk of bias item presented as percentages across all included studies. Green–Low risk of bias; Yellow–Unclear risk of bias; and Red–High risk of bias.

**Figure 5 diseases-11-00138-f005:**
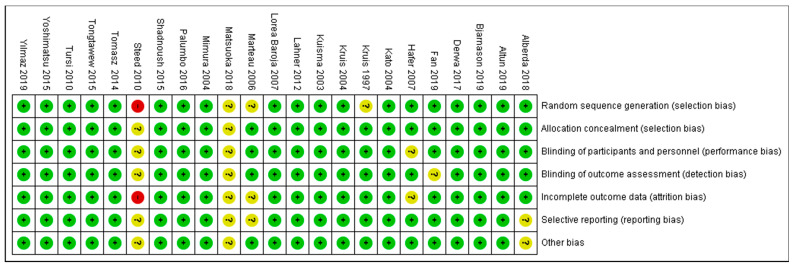
Risk of bias summary [21,22,23,24,25,27,28,30,31,33,34,35,36,37,39,41,43,45,46,47,48,50,51].

**Figure 6 diseases-11-00138-f006:**
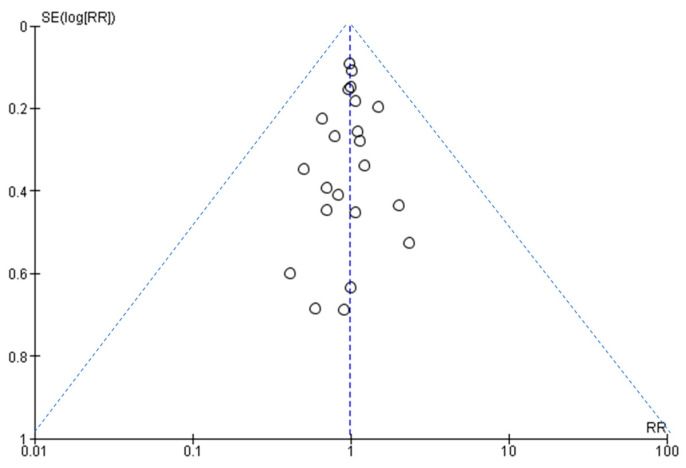
A funnel plot to assess publication bias.

**Figure 7 diseases-11-00138-f007:**
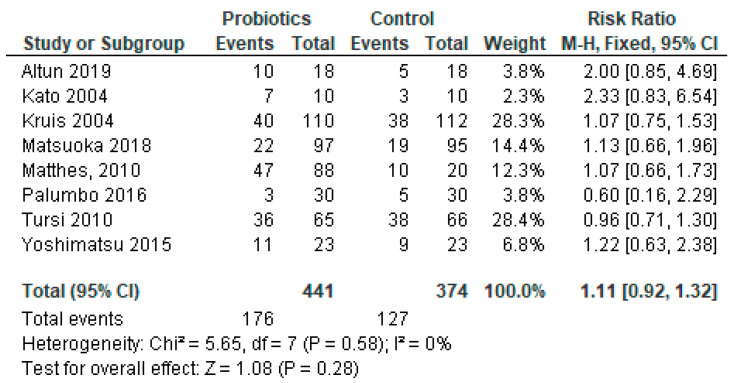
Pooled analysis of 8 studies reporting patients with ulcerative colitis infection only [22,28,30,37,38,41,48,51].

**Table 1 diseases-11-00138-t001:** Type of gastroenteritis, probiotics used and their effectiveness.

Authors, Year [Ref.]	Disease Conditions ^1^ and Sample Size	Type of Probiotics Used	Prevention/Treatment	Effective	Quality Appraisal and Scoring ^6^	Major Findings/Conclusions	Country
Alberda et al., 2018 [21]	Antibiotic-associated diarrhea (AAD) and *Clostridium difficile*-induced diarrhea; sample size = 32	*Lactobacillus casei*	Prevention	Yes	Score 3 out of 4 (Moderate)	Probiotic drinks can prevent AAD and *Clostridium difficile* infections.	Canada
Altun, et al., 2019 [22]	Inflammatory bowel disease (IBD) ^2^: ulcerative colitis (UC); sample size = 40	*Enterococcus faecium*, *Lactobacillus plantarum*, *Streptococcus thermophilus*, *Bifidobacterium lactis*, *Lactobacillus acidophilus*, *Bifidobacterium longum*)-and fructooligosaccharide	Treatment	No	Score 4 out of 4 (High)	The use of synbiotic therapy^3^ had no statistically significant effect in the improvement of clinical and endoscopic parameters compared with controls.	Turkey
Bjarnason, et al., 2019 [23]	IBDs: UC (*n* = 81) and Crohn’s disease (CD) (*n* = 61); total samples more than 500.	*Symprove contains multiple strains of probiotics such as Lactobacillus plantarum*, *Lactobacillus rhamnosus*, *Lactobacillus plantarum*, *Lactobacillus acidophilus*, *and E. faecium*	Treatment	Yes	Score 4 out of 4 (High)	Multi-strain probiotics decreased intestinal inflammation in patients with UC, but not in patients with CD.	United Kingdom
Derwa, et al., 2017 [24]	IBDs ^2^; sample size = 777	*Probiotics* vs. 5-aminosalicylates (5-ASAs) (in one RCT); probiotics vs. placebo (in 7 RCTs).	Treatment	No	Score 4 out of 4 (High)	There was no benefit of probiotics over 5-ASAs ^4^ or placebo in inducing remission in active inflammatory bowel diseases. For UC, relative risk of failure to achieve remission = 0.86; 95% CI = 0.68–1.08.	Multiple countries
Fan, et al., 2019 [25]	IBDs ^2^; sample size = 40	Bifico contains probiotic bacteria Bacillus Coagulans GBI-30, 6086. Bifico was given as an adjuvant treatment with Pentasa, which is an anti-inflammatory agent.	Treatment	Yes	Score 4 out of 4 (High)	A combination of probiotics and pentasa can improve microflora composition in patients with IBD and reduce the level of inflammatory cytokines.	China
Groeger, et al., 2013 [26]	IBDs ^2^; chronic fatigue syndrome (CFS); Psoriasis. Sample sizes: UC = 22, CFS = 48, psoriasis = 26.	*Bifidobacterium infantis* 35624	Treatment	Yes	Score 4 out of 4 (High)	Microbiota in humans have immuno-modulatory effects on both the mucosal and systemic immune systems.	Ireland
Hafer, et al., 2007 [27]	IBDs ^2^; sample sizes: UC = 14, Crohn’s disease = 17.	Standard treatment vs. standard treatment with oral lactulose.	Treatment	No	Score 3 out of 4 (Moderate)	Oral lactulose has no beneficial no clinical and immunological effects on IBD patients.	Germany
Kato, et al., 2004 [28]	IBD: UC; sample size = 20	*Bifidobacteria-fermented milk (BFM)*	Treatment	Yes	Score 3 out of 4 (Moderate)	Supplementation with BFM has beneficial effects in managing active ulcerative colitis and is more effective than the conventional treatment alone.	Japan
Krag, et al., 2012 [29]	IBD: UC; sample size = 39	Profermin, consisting of fermented oats, *Lactobacillus plantarum 299v*, *barley malt*, *lecithin*, *and water*	Treatment	Yes	Score 4 out of 4 (High)	Profermin is safe and may be effective in inducing remission of active ulcerative colitis.	Denmark
Kruis, et al., 1997 [30]	IBD: UC; sample size = 120	*Escherichia coli* Nissle (Serotype 06: K5: H1), as an adjuvant treatment with mesalazine (also known as 5-aminosalicylic acid (5-ASA) ^5^	Treatment	Yes	Score 4 out of 4 (High)	*E. coli* (Serotype 06: K5: H1) effectively prevents remission of ulcerative colitis as a standard treatment with 5-ASA ^4^.	Germany, Czech Republic, and Austria
Kruis, et al., 2004 [31]	IBD: UC; sample size = 327	*Escherichia coli* Nissle 1917	Maintaining remission and prevention of relapses	Yes	Score 4 out of 4 (High)	Probiotic EcN has therapeutic effects and is safe for remission in UC. EcN can be used as an alternative to 5-ASA ^4^.	Germany
Kuehbache, et al., 2006 [32]	Pouchitis ^4^; sample size = 15	*VSL #3* consists of *Lactobacillus casei*, *L. plantarum*, *L. acidophilus*, *L. bulgaricus*, *Bifidobacterium longuum*, *B. breve*, *B. infantis*, *and Streptococcus salivarius* sub-spp. *Thermophillus*	Treatment	Yes	Score 3 out of 4 (Moderate)	Probiotic therapy with *VSL #3 increases the diversity*, *richness and total number of intestinal bacteria and bacterial microbiota.*	Germany
Kuisma, et al., 2003 [33]	Pouchitis ^4^; sample size = 20	*Lactobacillus rhamnosus GG*	Treatment	No	Score 3 out of 4 (Moderate)	*Lactobacillus GG* can alter the microbial flora in ileo-anal pouches but was inefficient for clinically improving pouch inflammation.	Finland
Lahner, et al., 2012 [34]	Symptomatic uncomplicated diverticular disease; sample size = 45	*Lactobacillus paracasei* B21060 (symbiotic sachet Flortec^©^) plus high fiber diet (Treatment Group) vs high fiber diet only (Controls)	Treatment	Yes	Score 3 out of 4 (Moderate)	The treatment group having symbiotic sachet Flortec^©^ plus a high fiber diet improved of clinical symptoms (abdominal pain, bloating) significantly more than the control group.	Italy
Lorea Baroja, et al., 2007 [35]	IBD: Crohn’s disease (*n* = 15) and UC (*n* = 5), control, (*n* = 20); total sample size = 40	*Lactobacillus rhamnosus GR-1 and L. reuteri RC-14- supplemented yogurt* vs. *placebo*	Prevention	Yes	Score 4 out of 4 (High)	Short-term probiotic yogurt consumption with *Lactobacillus rhamnosus GR-1 and RC-14* has beneficial immune modulatory effects.	Canada
Marteau, et al., 2006 [36]	IBD: Crohn’s disease; sample size = 98	*Lactobacillus johnsonii LA1*	Prevention of relapses	No	Score 4 out of 4 (High)	*Lactobacillus johnsonii LA1 had no sufficient effect to prevent the recurrence of Crohn’s disease.*	France
Matsuoka, et al., 2018 [37]	IBD: UC; sample size = 195	*Bifidobacterium breve* fermented milk (BFM)	Prevention	No	Score 4 out of 4 (High)	BFM had no effect on the time to relapse in UC patients, compared with placebo.	Japan
Matthes, et al., 2010 [38]	IBD: UC; sample size = 90 (70 with UC and 20 controls)	*Escherichia coli* Nissle 1917 (EcN)	Treatment	Yes	Score 3 out of 4 (Moderate)	*Escherichia coli* Nissle 1917 (EcN) may be an alternative treatment for moderate distal UC.	Germany
Mimura, et al., 2004 [39]	Recurrent or refractory pouchitis ^3^; sample size = 36	VSL #3 contains *Lactobacillus casei*, *L. plantarum*, *L. acidophilus*, *L. bulgaricus*, *Bifidobacterium. longuum*, *B. breve*, *B. infantis Streptococcus salivarius* subsp. *Thermophillus*	Treatment of remission	Yes	Score 3 out of 4 (Moderate)	VSL#3 probiotic therapy is highly effective in maintaining treatment of recurrent pouchitis ^3^ and improving quality of life.	United Kingdom and Italy
Mitra & Rabbani, 1990 [14]	Acute watery diarrhea due to *Vibrio cholerae* and *E. coli* infection; sample size = 183	Bioflorin (*Streptococcus faecium* SF68), given orally along with intravenous rehydration, and followed by oral rehydration solution	Treatment	No	Score 4 out of 4 (High)	Bioflorin was not effective in treating acute diarrhea due to *V. cholerae* and enterotoxigenic *E. coli* infections.	Bangladesh
Montalto, et al., 2010 [40]	Non-steroidal anti-inflammatory drug-induced enteropathy; sample size = 20	VSL #3 contains *Lactobacillus casei*, *L. plantarum*, *L. acidophilus*, *L. bulgaricus*, *Bifidobacterium. longuum*, *B. breve*, *B. infantis Streptococcus salivarius* subsp. *Thermophillus*	Treatment	Yes	Score 2 out of 4 (Poor)	Probiotics mixture could be useful in decreasing indomethacin-induced intestinal inflammation.	Italy
Palumbo, et al., 2016 [41]	IBD: UC; sample size = 60	A probiotic blend, which consists of *Lactobacillus salivarius*, *Lactobacillus acidophilus* and *Bifidobacterium bifidus* strain BGN4, given as an adjuvant therapy with Mesalazine	Treatment	Yes	Score 3 out of 4 (Moderate)	Long-term treatment modality of anti-inflammatory drugs and probiotics is viable and could be an alternative treatment for mild-to moderate UC.	Italy
Persborn, et al., 2013 [42]	Pouchitis ^4^; sample size = 16 patients with pouchitis and 13 controls with a healthy ileoanal pouch	*Bifidobacterium bifidum (W23)*, *B. lactis (W51)*, *B. lactis (W52)*, *Lactobacillus acidophilus (W22)*, *L. casei (W56)*, *L. paracasei (W20)*, *L. plantarum (W62)*, *L. salivarius (W24)*, *L. lactis (W19)*	Treatment	Yes	Score 3 out of 4 (Moderate)	Probiotics restored the mucosal barrier to *E. coli* in patients with pouchitis ^3^. This can prevent recurrence during maintenance therapy.	Sweden
Shadnoush, et al., 2015 [43]	IBDs ^2^; sample size = 305, of which 105 IBD patients received probiotic yogurt, 105 IBD patients received placebo, and 95 healthy controls received probiotic yogurt	Probiotic yogurt containing *Lactobacillus acidophilus La-5 and Bifidobacterium BB-12*	Treatment	Inconclusive	Score 3 out of 4 (Moderate)	Fiber and energy intake in the treatment group did not increase when compared with those of controls. However, consumption of probiotic yogurt by patients with IBD may help to increase the number of probiotic bacteria in the intestine, thus improving intestinal function.	Iran
Shen, et al., 2005 [44]	Antibiotic-dependent pouchitis ^4^; sample size = 31	VSL #3 contains four strains of *Lactobacillus*, three *Bifidobacterium* species, *Streptococcus salivarius* subsp. *thermophillus*	Treatment	Yes	Score 3 out of 4 (Moderate)	The use of probiotics is useful, and the authors suggested it in routine clinical care.	United States
Steed, et al., 2010 [45]	IBD: Crohn’s disease; sample size = 35	*Bifidobacterium longum* and Synergy 1 which contains Orafti, Tienen, Belgium	Treatment	Yes	Score 4 out of 4 (High)	Effective in improving clinical symptoms in patients with active Crohn’s disease.	Scotland
Tomasz, et al., 2014 [46]	Pouchitis ^4^; sample size = 43	*Lactobacillus acidophillus*, *L. delbrueckii subsp. bulgaricus*, and *Bifidobacterium bifidus*	Prevention	Yes	Score 4 out of 4 (High)	Long-term use of probiotics is safe and can be an effective method of preventing pouchitis ^3^.	Poland
Tongtawee, et al., 2015 [47]	*Helicobacter pylori*; sample size = 200	*Lactobacillus delbrueckii subsp. bulgaricus* and *Streptococcus thermophillus*	Treatment	Yes	Score 4 out of 4 (High)	Pretreatment with probiotic containing yogurt can potentiate the effects of triple therapy for *Helicobacter pylori*.	Italy
Tursi, et al., 2010 [48]	IBD: UC; sample size = 144	*VSL #3* consists of *Lactobacillus casei*, *L. plantarum*, *L. acidophilus*, *L. bulgaricus*, *Bifidobacterium longuum*, *B. breve*, *B. infantis*, *and Streptococcus salivarius* sub-spp. *Thermophillus*	Treatment	Yes	Score 4 out of 4 (High)	High potency probiotic mixture supplementation is safe and improves rectal bleeding and reduce remission in relapsing UC patients after 8 weeks of treatment.	Thailand
Venturi, et al., 1999 [49]	IBD: UC; sample size = 20	*VSL #3* consists of *Lactobacillus casei*, *L. plantarum*, *L. acidophilus*, *L. bulgaricus*, *Bifidobacterium longuum*, *B. breve*, *B. infantis*, *and Streptococcus salivarius* sub-spp. *Thermophillus*	Treatment	Yes	Score 3 out of 4 (Moderate)	Intake of VSL #3 preparation enhances the concentrations of some strains of protective bacteria in the intestinal microflora.	Italy
Yilmaz, et al., 2019 [50]	IBDs ^2^; sample size = 45	Kefir, a cultured, fermented beverage, which contains *Lactobacillus* bacteria	Treatment	Yes	Score 3 out of 4 (Moderate)	Consumptions of kefir has short term effects on improving the quality of life of patients.	Turkey
Yoshimatsu, et al., 2015 [51]	IBD: UC; sample size = 46	*Streptococcus faecalis (lactomin)*, *Clostridium butyricum*, *and Bacillusmesentericus*	Prevention of relapse	Yes	Score 3 out of 4 (Moderate)	Probiotics may be effective for maintaining clinical remission in patients with quiescent UC.	Japan
Ziemniak, 2006 [52]	Chronic gastritis, or duodenal ulcer caused by *Helicobacter pylori*; sample size = 641	Lacidofil containing *Lactobacillus acidophilus and Lactobacillus rhamnosus*, as an adjuvant therapy with antibiotics and proton pump inhibitor (PPI)	Treatment	Yes	Score 4 out of 4 (High)	Lacidofil increases the efficacy of clarithromycin and amoxicillin and also reduces complications of antibiotic therapy.	Poland
Zocco, et al., 2006 [53]	IBD: UC; sample size = 187	*Lactobacillus GG*	Treatment and prevention of remissions	Yes	Score 4 out of 4 (High)	*Lactobacillus GG* is effective and safe for maintaining ulcerative colitis remission and could be a good therapeutic alternative.	*Italy*
Zwolinsk, et al., 2009 [54]	IBD: UC; sample size = 101, of which 56 had active phase of UC, 33 non-active phase of UC, and 12 IBS controls	*Lacidofil*, containing two well-characterized strains of Lactobacillus: *L. helveticus* R-52 and *L. rhamnosus* R-11.	Treatment	Yes	Score 4 out of 4 (High)	Probiotic therapy is beneficial in counteracting the effects of delayed healing of trinitrobenzene sulfonic acid induced colitis caused by *Candida*.	Poland

^1^ Disease conditions included: inflammatory bowel diseases (IBDs), pauchitis, antibiotic-associated diarrhea, *Helicobacter pylori* infection, diverticulitis, and acute watery diarrhea; ^2^ Inflammatory bowel diseases (IBD) include ulcerative colitis (UC) and Crohn’s disease (CD); ^3^ Synbiotic therapy: a combination of prebiotics and probiotics; ^4^ Pouchitis: inflammation of a J-shaped pouch, which is created by surgical procedures as a treatment of UC; ^5^ 5-ASA: 5-aminosalicylates; mesalazine is a 5-ASA; ^6^ the scoring system included the following criteria: (1) Sample size: greater than 30 = 1; smaller sample = 0; (2) Randomized controlled trials = 1; not randomized, no controls = 0; (3) Studied both safety and efficacy = 1; otherwise = 0; (4) Standard and objective evaluation criteria = 1, otherwise = 0.

**Table 2 diseases-11-00138-t002:** Effectiveness of probiotics, as evaluated through meta-analyses.

	Probiotics		Control Group			Risk Ratio
Study	Events	Total	Events	Total	Weight (%)	M-H, Random, 95% CI
Alberda et al., 2018 [21]	11	16	10	16	3.7	1.10 [0.67, 1.82]
Altun et al., 2019 [22]	10	18	5	18	1.3	2.00 [0.85, 4.69]
Bjarnason et al. 2019 [23]	4	77	4	70	0.5	0.91 [0.24, 3.50]
Fan et al., 2019 [25]	7	21	9	19	1.6	0.70 [0.33, 1.52]
Hafer et al., 2007 [27]	6	15	6	16	1.2	1.07 [0.44, 2.59]
Kato et al., 2004 [28]	7	10	3	10	0.9	2.33 [0.83, 6.54]
Kruis et al., 1997 [30]	18	50	29	53	4.7	0.66 [0.42, 1.03]
Kruis et al., 2004 [31]	40	110	38	112	7.0	1.07 [0.75, 1.53]
Kuisma et al., 2003 [33]	9	10	9	10	10.0	1.00 [0.75, 1.34]
Lahner et al., 2012 [34]	27	30	20	22	21.8	0.99 [0.83, 1.18]
Lorea Baroja et al., 2007 [35]	4	20	4	20	0.6	1.00 [0.29, 3.45]
Marteau et al., 2006 [36]	30	47	21	49	6.0	1.49 [1.01, 2.20]
Matsuoka et al., 2018 [37]	22	97	19	95	3.2	1.13 [0.66, 1.96]
Miruma et al., 2004 [39]	7	20	11	16	2.1	0.51 [0.26, 1.01]
Palumbo et al., 2016 [41]	3	30	5	30	0.5	0.60 [0.16, 2.29]
Shadnoush et al., 2015 [43]	30	176	18	84	3.4	0.80 [0.47, 1.34]
Steed et al., 2010 [45]	5	13	6	11	1.3	0.71 [0.29, 1.69]
Tomasz et al., 2014 [46]	3	19	8	21	0.7	0.41 [0.13, 1.34]
Tongtawee et al., 2015 [47]	62	98	60	96	16.3	1.01 [0.82, 1.26]
Tursi et al., 2010 [48]	36	65	38	66	9.5	0.96 [0.71, 1.30]
Yilmaz et al., 2019 [50]	11	23	9	23	2.2	1.22 [0.63, 2.38]
Yoshimatsu et al., 2015 [51]	5	10	6	10	1.5	0.83 [0.37, 1.85]
**Total (95% CI)**		**975**		**867**	**100%**	**0.99 [0.90, 1.09]**
Total events	357		338			
Heterogeneity: Tau^2^ = 0.00; Chi^2^ = 22.43, df = 21 (*p* = 0.37); I^2^ = 6%						
Test of overall effect Z = 0.18 (*p* = 0.85)						

## Data Availability

Research data included publicly available data using data sources such as PubMed, Google Scholar, MEDLINE and Scopus. In this study, only clinical trials were included.

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
