# Peer review of "Effects of Probiotics in Adults with Gastroenteritis: A Systematic Review and Meta-Analysis of Clinical Trials"

_diseases, 2023, doi:10.3390/diseases11040138_

Round 1
Reviewer 1 Report
Reviewer comments and suggestions
The study aimed to review existing literature on the effects of probiotics in gastroenteritis among adults. The authors only used original articles on clinical trials that demonstrated the effects of probiotics in adults with gastroenteritis Thirty-five articles met the selection criteria; of them, probiotics were found effective in the treatment and/or prevention of chronic inflammatory bowel disease (IBD) including ulcerative colitis and Crohn’s disease in 17 (49%), and the treatment of pouchitis in 4 (11.4%), antibiotic-induced diarrhea in 3 (8.6%), Helicobacter pylori infection in 2 (5.7%) and diverticulitis in 1 (2.9%), while the remaining 7 (20%) were ineffective and 1 study results were inconclusive. The study concluded that meta-analysis didn’t demonstrate any significant protective effects of probiotics.
Overall, the manuscript was well written. However, a few concerns/comments needed to be explained/modified.
- Line 39 Need to add more on the symptoms
- Line 78-79 Can you please explain more about the number of the manuscript that were excluded
- Comments for table 1 is only this type of gastroenteritis was present?
- In the legend part of Table 1 please explain the information of scoring system used.
- Line 139-140 Which studies were talking here that probiotics are ineffective?
- Line 146 Were the disorders were same in all studies or different such as IBD, as shown in Figure 2
- Line 148-149 How the authors explain the positive results obtained from various research in animal and human studies.
- Line 195-196 The Authors could categorize various diseases into individuals and then try to do the analysis, I think they can get some positive
- Line 232 The authors did not find recent studies to be cited
- Line 251 I think it would be not a limitations
- ‘Please check the reference 50 and change according to MDPI guidelines
Reviewer 2 Report
The manuscript is well-structured and effectively addresses the topic of probiotics in gastroenteritis treatment. It demonstrates a thorough and systematic approach to reviewing the existing literature. While the manuscript has several strengths, there are also some areas where minor improvements could enhance its quality.
The introduction is quite dense and could benefit from more concise and clear wording. The introduction jumps between various aspects of gastroenteritis, its global impact, probiotics, and the study's purpose. It could benefit from a more organized structure that gradually leads the reader to the study's objective. When mentioning the potential benefits of probiotics, it would be helpful to provide specific examples or findings from previous research. This would add depth to the introduction.
Overall, the "Materials and Methods" section effectively outlines the steps taken in conducting the systematic literature search and quality appraisal. Providing a bit more detail in certain areas and explaining the rationale behind criteria could further enhance its clarity and completeness. Specifying how the Boolean operators (AND, OR) were used could provide a more comprehensive understanding of the search process. The inclusion criteria for studies are clearly outlined, which is essential for maintaining transparency and reproducibility. However, stating the rationale behind selecting the age range of participants (19-55 years) would enhance the comprehensibility of the criteria. The explanation of using CADIMA for quality appraisal and the rating scale (0-4) is clear and informative. However, some readers may benefit from a brief explanation of the CADIMA tool and its key features, especially if they are unfamiliar with it.
The "Results" section provides a clear overview of the screening process and key findings The subsection provides valuable information about the types of probiotic strains used and their administration in the studies. The description of follow-up protocols is clear, but it might be useful to provide a summary or comparison of the follow-up durations for different illnesses.
Overall, the "Discussion" section provides a thorough analysis of the study's findings, strengths, and limitations. It effectively communicates the complexities surrounding the effectiveness of probiotics in treating gastroenteritis in adults and the need for additional research. However, it could benefit from more discussion on the implications of these findings and potential future research directions. Additionally, the section might consider addressing the potential limitations of the studies included in the meta-analysis in more detail.
Reviewer 3 Report
Many studies are available on the effectiveness of probiotic treatment of childhood gastroenteritis. However, much limited number of adult studies are ready to hand, and their results are not unanimously conclusive. Therefore, the subject of the present manuscript is very actual.
The authors give a profound analysis based on a large number of studies performed in various countries representing different continents, and apply strict exclusion criteria.
Some questions and remarks:
Why was 59 years taken as the upper age limit? Though the onset of IBDs comes generally at an earlier age, their prevalence is significant even above 60.
There is no such age limit in many of the referenced papers. Furthermore, sex, racial and ethnical proportions, and co-morbidities can be revealed only from the papers referenced. This is true also for the types of antibiotics applied for prevention of antibiotic associated diarrhoea. Neither of these parameters are treated in Table 1.
When a probiotic is applied for the prevention of antibiotic associated diarrhoea can it exactly be known how many diarrhoea would have occurred without the treatment? In other words, can it be declared that more cases would have occurred in the treated persons without treatment?
Round 2
Reviewer 3 Report
No further comments.